# Metal Oxide Nanowires Grown by a Vapor–Liquid–Solid Growth Mechanism for Resistive Gas-Sensing Applications: An Overview

**DOI:** 10.3390/ma16186233

**Published:** 2023-09-15

**Authors:** Ali Mirzaei, Myoung Hoon Lee, Krishna K. Pawar, Somalapura Prakasha Bharath, Tae-Un Kim, Jin-Young Kim, Sang Sub Kim, Hyoun Woo Kim

**Affiliations:** 1Department of Materials Science and Engineering, Shiraz University of Technology, Shiraz 71557-13876, Iran; mirzaei@sutech.ac.ir; 2Department of Materials Science and Engineering, Inha University, Incheon 22212, Republic of Korea; dm249@naver.com (M.H.L.); 1krishnaisk@gmail.com (K.K.P.); pbharathbhat@gmail.com (S.P.B.); xodjs635@naver.com (T.-U.K.); piadote@naver.com (J.-Y.K.); 3Division of Materials Science and Engineering, Hanyang University, Seoul 04763, Republic of Korea; 4The Research Institute of Industrial Science, Hanyang University, Seoul 04763, Republic of Korea

**Keywords:** metal oxide, nanowire, VLS mechanism, gas sensor, sensing mechanism

## Abstract

Metal oxide nanowires (NWs) with a high surface area, ease of fabrication, and precise control over diameter and chemical composition are among the best candidates for the realization of resistive gas sensors. Among the different techniques used for the synthesis of materials with NW morphology, approaches based on the vapor–liquid–solid (VLS) mechanism are very popular due to the ease of synthesis, low price of starting materials, and possibility of branching. In this review article, we discuss the gas-sensing features of metal oxide NWs grown by the VLS mechanism, with emphasis on the growth conditions and sensing mechanism. The growth and sensing performance of SnO_2_, ZnO, In_2_O_3_, NiO, CuO, and WO_3_ materials with NW morphology are discussed. The effects of the catalyst type, growth temperature, and other variables on the morphology and gas-sensing performance of NWs are discussed.

## 1. Introduction

One-dimensional (1D) materials such as nanofibers (NFs), nanorods (NRs), nanotubes, and nanowires (NWs) have two dimensions in the range of 1–100 nm, but one dimension is much longer [1]. In particular, NWs have lengths in the range of micrometers and high aspect ratios of 20 or higher. Their cross-sections are usually round [2,3,4]. NWs generally have high crystalline quality, have tunable composition and properties, and, due to their morphology, they can also be incorporated in different devices with different designs [5,6].

In general, “top-down” and “bottom-up” synthesis methods are employed for the synthesis of nanostructures [7]. The former methods such as lithography techniques use bulk materials; by sculpting or etching them, nanostructures can be produced [8,9]. However, due to nanoscale dimensions, in some cases, the quality of the nanostructures cannot be adequately controlled [2]. In bottom-up approaches, controlled crystallization from vapor or liquid sources is possible, which leads to the realization of high-quality nanostructures [2]. Based on the above approaches, there are different methods available to synthesize metal oxide NWs including hydrothermal [10], sol–gel [11], sonochemical [12], lithography [13], confinement growth [14], pulse laser deposition (PLD) [15], direct thermal oxidation [16], and chemical vapor deposition (CVD) [17].

In 1964, Wagner [18] presented a vapor–liquid–solid (VLS) mechanism for the synthesis of Si whiskers on a Au-deposited Si substrate using SiCl_4_ and H_2_ source materials. By heating above 363 °C, which is above the eutectic point of Au-Si, a Au-Si liquid droplet can be formed. Subsequently, additional Si in the vapor phase is readily adsorbed on the droplet, increasing the Si amount in the droplet. After being supersaturated with Si atoms, the driving force for the crystallization of excess Si at the liquid–solid interface is provided, and Si whiskers grow. After becoming fully grown, the droplet remains on the tip of the whiskers through surface tension [19,20]. Later, Holonyak and co-workers [21] reported the synthesis of GaP whiskers using a Zn catalyst. Subsequent research by Givargizov [22] and others further explained the details of the VLS mechanism. In general, any unidirectional growth of crystalline materials with the help of a liquid phase and sources provided from a vapor phase is related to the VLS growth mechanism [23]. Different materials such as metal oxides [24] and metal sulfides [25] are currently synthesized by this mechanism via different techniques [26,27].

The advantages of this mechanism include (i) the synthesis of good quality crystalline 1D nanostructures such as NWs, (ii) the possibility of tuning chemical compositions, (iii) a simple setup, and (iv) low activation energy compared to other vapor deposition approaches. Nevertheless, sometimes a high growth temperature is a limiting factor for approaches which use this technique [28]. Moreover, this approach cannot be used for the growth of metallic NWs; for some applications, the existence of a metal catalyst at the tip of nanostructures may degrade its properties [29]. However, for gas-sensing purposes, the presence of a metal catalyst, which is usually a noble metal, enhances the sensing response by chemical sensitization and electronic sensitization. It should be noted that today, mostly metal oxides with a particulate morphology are used for the realization of gas sensors. They have a high surface area, and their contact points are considered good sources of resistance modulation. However, extensive agglomeration often occurs among them, leading to an extensive decrease in their surface area and closing of open pores among them. Hence, the sensing performance is decreased, especially when the sensor is working at a high temperature due to agglomeration among particles. An alternative way is to use the metal oxides with NW morphology. Metal oxide NWs have stable chemical properties as well as a large surface area. In particular, when the diameter of NWs approaches their Debye lengths, significant modulation of the resistance can occur in the presence of target gases. Moreover, NWs can have large mechanical deformations without cracking, being so small that they are incredibly robust and tough. Hence, peculiar assemblies can be achieved on flexible substrates to develop high-performance flexible/wearable gas sensors. Furthermore, it is reported that NPs have no efficient self-heating effects, which typically form agglomerations. In contrast, NWs with 1D morphology have good efficiency in self-heating mode [30,31,32,33,34]. Therefore, it seems that metal oxide NWs can be promising materials for the realization of highly sensitive, flexible, and low-power-consumption gas sensors.

## 2. VLS Growth Mechanism and Fundamentals

Often, the VLS mechanism has four steps (Figure 1): (i) the transport of vapor phase precursors, (ii) adsorption and dissolution of vapors at the droplet, resulting in catalyst alloy formation, (iii) diffusion of dissolved species in droplets, and (iv) precipitation and growth at the liquid–solid (L/S) interface.

Catalyst alloy formation for two- and three-valence metal oxides such as ZnO and In_2_O_3_ can be described as follows [36]. From MDPI:M(v) + Au(l) = AuM(l) for two-valence metal oxides(1)
2M(v) + 2Au(l) = 2AuM(l) for three-valence metal oxides(2)

The formation of NWs proceeds as follows [36]:AuM(l) + M(v) + ½ O_2_ = AuM(l) + MO(s) for two-valence metal oxides(3)
2AuM(l) + 4M(v) + 3O_2_ (g) = 2AuM(l)+ 2M_2_O_3_(s) for three-valence metal oxides(4)

Figure 2 schematically shows a typical thermal CVD setup for the synthesis of NWs using a VLS mechanism [37]. The process is usually performed in a system composed of a horizontal tube furnace, a pump system, an alumina or quartz tube, a gas supply, and a gas control unit. The carrier gas enters the inlet of the tube. The starting material is put on a crucible, and then it is put at the center of the tube, in which the maximum temperature is provided. The substrate at a lower temperature is placed downstream at a fixed distance from the alumina boat for collecting growth products. Upon evacuating the tube, thermal vaporization occurs, and growth starts on the substrate [38,39].

In more detail, usually, clean and macro-defect-free Si substrates are used for VLS growth. However, sometimes alumina or glass are also used. First, a layer of catalyst, usually Au, is coated over the substrate using different techniques such as sputtering. Subsequently, the substrate is put in a reaction tube, and it is heated where the catalyst is melted. Depending on the type of catalyst and substrate, the droplet starts to melt at a certain temperature. For Si substrates, by heating above the eutectic point of 363 °C in the Si-Au system, a droplet will appear. Then, the vapor-phase gas of the desired material to be grown flows through the reaction tube. The starting material can be in a solid form, and by using different heating techniques, it will evaporate. Vapor-phase atoms reach the droplet surface using diffusion or direct impingement. Indeed, due to the larger sticking coefficient of the liquid droplet relative to that of the solid substrate, the vapor-phase atoms prefer to be deposited onto the droplet and form an alloy. When the droplet is supersaturated with vapor-phase atoms, the driving force for the growth is provided, and crystal growth occurs at the L/S interface by precipitation where unidirectional growth starts, leading to the growth of NWs. During the growth of metal oxides, Ar or N_2_ as the carrier gas and oxygen gas are used. O_2_ gas can affect the stoichiometry and formation of metal oxide NWs. The NW diameter is mainly affected by droplets sizes. Moreover, the growth rate mostly depends on supersaturation, which is governed by the amount of the vapor-phase precursor and the substrate temperature. Often, the droplet is attached at the tip of the NW [40]. Hence, 1D nanostructures with a high aspect ratio and vertically aligned NWs can be synthesized [41,42].

When there is no catalyst for the growth of nanostructures, it is known as vapor–solid (VS), and growth occurs in a random distribution on the substrate. Moreover, the motivation for growth is the decrease in surface free energy. Upon the first condensation, the as-formed seeds serve as nucleation sites for subsequent growth. Actually, they facilitate directional growth to lower the surface energy [33]. However, the nanostructure produced by the VLS mechanism shows much higher aspect ratios and better crystalline quality relative to those grown by the VS mechanism [43]. Different synthesis techniques including physical routes such as PLD, thermal evaporation, and chemical routes such as CVD are used to provide the vapor-phase atoms for the growth of NWs and to tune the crystallinity, diameter, and aspect ratio in the VLS growth mechanism [33,44].

### 2.1. Metal Catalyst

Since the growth, growth site, and diameter of NWs are mainly governed by catalysts, catalysts are essential during VLS growth [45]. A liquid alloy is usually formed with the material within its growth conditions. A good metal catalyst should have these merits: (i) The precursor should be soluble in the catalyst to supersaturate it, and (ii) the catalyst should not have a low surface tension due to the lack of stability. Furthermore, it causes the formation of flatter droplets, which are not suitable for VLS growth [2]. (iii) The vapor pressure of the catalyst should be small, and (iv) it must be chemically inert [46]. Generally, Au is commonly used as a catalyst since it is inert and can be easily coated onto a substrate while it forms a eutectic with most substrates [47]. However, sometimes other catalysts are also used. For example, ultrathin (15 nm) GeS NWs have been grown in the presence of Bi as a catalyst [48].

### 2.2. Effect of Deposition Temperature

The selection of the source temperature depends on the volatile nature of the source material. However, it is slightly lower than the melting point (T_m_) of the starting material. The temperature can affect the droplet sizes, the amount of the generated vapors, and the length of NWs. Generally, higher deposition temperatures lead to smaller and denser droplets, resulting in the formation of thinner and denser NWs. At lower temperatures, the NWs become thicker and shorter. Indeed, at higher temperatures, more vapors and faster diffusion lead to the formation of denser, thinner, and longer NWs. However, very high temperatures lead to a high desorption rate, lessening the crystal growth [49].

### 2.3. Effect of the Deposition Pressure

As the pressure inside the chamber decreases, the generated vapors are transferred to be adsorbed on the Au droplets in the lower-temperature region, leading to crystal growth. The growth rate drops as the pressure decreases because less of the vapors can be moved to the lower temperature zone, leading to a slower growth rate [13].

### 2.4. Effect of the Oxygen Gas Flow

Metal vapors are oxidized more at a higher oxygen flow rate. Hence, the growth rate often increases with the oxygen flow rate. However, it should be noted that a very high oxygen flow rate induces the growth of NWs that are too long [13]. In the remaining parts of this review paper, we discuss the gas-sensing properties of metal oxide NWs grown by the VLS growth mechanism. It should be noted that there are already some review papers discussing VLS growth of different materials for different applications. However, as far as we know, there is no review paper related to VLS-grown metal oxide NWs for gas-sensing applications.

## 3. VLS-Grown Metal Oxide NWs for Gas-Sensing Studies

### 3.1. SnO_2_ NWs

The T_m_ of Sn is only 231.9 °C. As a result, the SnO_2_ NWs can be easily synthesized by the VLS growth mechanism [50,51]. Other morphologies such as SnO_2_ nanobelts have also been reported by using this mechanism [52]. In addition, doping of other elements through starting materials or through the catalyst is possible [53]. During VLS growth, doping can be performed, and Sb is a stable dopant which can be easily incorporated into the SnO_2_ lattice. In this context, Sb-doped SnO_2_ NWs were synthesized by the VLS growth mechanism using Sn and Sb powders. They were synthesized at 950 °C for 90 min on a Au-coated (20 nm) Si/SiO_2_ substrate. Figure 3a,b show SEM micrographs of as-grown NWs and a single NW with a Au NP on its tip, respectively, demonstrating the VLS growth mechanism. Moreover, Figure 3c,d exhibit the EDS spectra captured at the Au NP and at the NW surface, respectively [54], in which the presence of Sb as the dopant can be confirmed. Even though these NWs probably had good potential for gas sensing, their sensing performance was not studied. As a result, this system needs to be explored more in regard to this aspect in the future.

In other research, SnO_2_ NWs were grown on bare SnO_2_ (10 nm) and Au (10 nm)-coated SiO_2_/Si (100) substrates at 900 °C for 1 h. The rectangular cross-section was obtained for the NWs deposited onto pristine and Au-coated substrates, while the NWs grown on a SnO_2_-coated substrate look like nanoribbons. Based on the H_2_ gas-sensing results, the sensor fabricated using SnO_2_ NWs grown on the Au-coated substrate had a higher response than other gas sensors. It displayed a response of 11.5 to 1000 ppm H_2_ gas at 100 °C. The Au NPs on the tip of the NWs improved the conductivity of NWs and resulted in easy H_2_ gas dissociation. In fact, Au NPs are not only helpful for the VLS growth of NWs but also enhance the response of the sensor. Furthermore, the high surface area allowed more H_2_ gases to be adsorbed on NWs, contributing to the sensing signal [55].

Kim et al. [56] synthesized SnO_2_ NWs and investigated the effect of temperature (700–900 °C) on their morphologies and diameters in the presence of Bi as a catalyst. Most of the NWs which were synthesized at 700 °C (Figure 4a) had a polygonal-shaped NP on their tips (inset in Figure 4a). The diameters of the NWs and NPs were 30–120 nm and 60–180 nm, respectively. Moreover, in the products grown at 800 °C, both particle-ended NWs and particle-free NWs (Figure 4b and inset) were observed. Figure 4c and its inset show that NWs grown at 900 °C had no particles on their tips. As shown in Figure 4d, when a mixed powder of Bi/Sn was evaporated, liquid droplets containing Sn, Bi, and O were adsorbed on the substrate. After supersaturation with these elements, short SnO_2_ nuclei precipitated at the S/L interface. By constantly dissolving the mentioned elements onto the NPs, SnO_2_ NWs grew along with crystalline Bi_2_Sn_2_O_7_ NPs on their tips. The sample grown at 700 °C was used as an oxygen gas sensor; at 25 °C, it was able to detect oxygen gas with a stable signal. However, no selectivity study was reported.

Sometimes, SnO powder can be used instead of Sn powder for the growth of SnO_2_ NWs [57]. For example, Cakestani et al. [58] used commercial SnO powder rather than elemental Sn powder to better control the evaporation process. The dissociation of SnO only takes place at 800–1000 °C.
2SnO (s) → Sn (l) + SnO_2_ (s)(5)

Furthermore, in spite of pure liquid Sn, the mixture of SnO_2_ (s) and Sn (l) did not flow inside the crucible. The dissociation starts in the presence of the carrier gas (Ar). Then, Sn vapors are moved to the substrates and are condensed into liquid droplets. Crystallization into SnO_2_ NWs occurs in the presence of oxygen gas. However, sensing studies were not carried out. In another study, SnO_2_ NWs were grown at 800 °C (using Sn powder) and 980 °C (using SnO powder) for 30 min on Si substrates. Furthermore, hierarchical SnO_2_ nanostructures were synthesized by producing SnO_2_ NWs first, followed by subsequent SnO_2_ branching. FE-SEM images of different SnO_2_ nanostructures are provided in Figure 5a–d. According to the sensing results, the responses of hierarchical SnO_2_ nanostructures were higher than those of SnO_2_ NWs for LPG and NH_3_ gases (Figure 6a,b). The width of the EDL as well as the height of the barrier potentials change in the presence of the target gas. Improved sensing characteristics of SnO_2_ hierarchical nanostructures are related to (i) the porous nature due to the hierarchical structure, (ii) the existence of much more active sites, and (iii) the presence of more homojunctions barriers between SnO_2_-SnO_2_ NWs in hierarchical nanostructures (Figure 6c,d) [59].

Au-coated Si substrates on which trenches with widths of 10 to 80 μm were produced by a photolithographic process were used for the growth of networked SnO_2_ NWs at 900 °C for 15 min [60]. In general, as the trench width decreased, the number of junctions increased (Figure 7a–f). The sensor with a 10 μm wide trench displayed a response of 3 to 10 ppm NO_2_ at 300 °C. Indeed, both the densely packed NWs and small trench size limited the flow of Sn vapor during VLS growth, mainly resulting in the growth of NWs at the upper surface of the trench. This eventually led to a poor response to NO_2_ gas due to a shortage of enough NWs for sensing of the gas. For the sensor with the 20 μm wide trench, the response increased to 25. However, further increasing the trench width resulted in decreased performance. This sensor had the largest overlapped area among NWs, leading to a lot of homojunctions between the NWs and a significant modulation of the resistance in the NO_2_ gas medium. Hence, the trench width should be optimized to obtain maximum response for a gas sensor.

Branched NWs can offer a large surface area, as an advantage for sensing applications. In this regard, by initially using metallic Sn powders, SnO_2_ NWs were grown on a Au (10 nm)-Si substrate at 960 °C for 1 h. Then, In_2_O_3_ NWs branches were grown on the stem of SnO_2_ NWs at 860 °C for 1 h using a self-catalytic VLS process. At 300 °C, the branched NWs comprised of SnO_2_ and In_2_O_3_ revealed a response of 1.4 to 10 ppm CO gas. The sensing mechanism was related to active centers due to oxygen vacancy defects, which facilitated the adsorption of oxygen and enhanced the interaction with CO gas, and the formation of heterojunctions between In_2_O_3_ and SnO_2_ along with the large surface area of the branched NWs [61].

One of unique features of the VLS growth mechanism is that the growth can occur directly on the electrodes on the substrate. In a relevant study using Sn powders, SnO_2_ NWs were directly fabricated on an electrode substrate comprised of Pt (5000 Å), Ti (500 Å), and Au (200 Å) layers. Growth at 700–800 °C for 20 to 30 min was carried out in the presence of different flow rates of O_2_ (1 to 10 sccm). The diameters of the NWs synthesized under different conditions were 20–60 nm. The optimal sensor fabricated from the NWs with the smallest diameter displayed a response of 43 to 10 ppm NO_2_ gas with a response time (t_res_) of 38 s and a recovery time (t_rec_) of 25 s at 200 °C. The presence of plenty of homojunctions between SnO_2_-SnO_2_ NWs led to improved response performance [62]. In another research work, SnO_2_ NWs were directly grown on Au-coated Si having electrodes with a gap size of 5 µm and using Sn powders at 750 °C for 30 min. The sensor displayed a high response of 110 to 5 ppm NO_2_ gas at 200 °C with fast dynamics. The formation of an electron depletion layer (EDL) around each NW as well as the formation of potential barriers at the SnO_2_-SnO_2_ NW junctions contributed to the sensing signal [63]. In another research work, single-crystalline SnO_2_ NWs were directly synthesized on a SiO_2_/Si substrate having electrodes by the VLS growth mechanism at 900 °C. SnO_2_ NWs with diameters of 30–200 nm and lengths of several micrometers were synthesized. At 150 °C, the sensor revealed a response of 7 to 1000 ppm H_2_ [64]. However, no other sensing tests such as selectivity were reported.

In another work, SnO_2_ NWs were synthesized on an interdigitated electrode (Ti, Pt, and Au; 10, 100, and 3 nm, respectively) substrate patterned by the VLS mechanism at 900 °C for 10 min, where subsequent attachment of Co_3_O_4_ was carried out via a sol–gel route. The response of Co_3_O_4_ NP-attached SnO_2_ NWs to 50 ppm acetone gas at 300 °C was ~7 times more than that of the bare sensor. In fact, in contact areas, due to the flow of electrons from SnO_2_ to Co_3_O_4_, the initial conduction channel inside the SnO_2_ NWs decreased relative to pure SnO_2_ NWs and subsequent significant expansion of the conduction channel in an acetone atmosphere, resulting in a high response to acetone gas [65]. In a similar study, SnO_2_ NWs were initially grown on an electrode substrate with a very similar procedure described previously [65]. Then, Fe_2_O_3_ NPs were coated onto SnO_2_ NWs by spin coating of a Fe_3_^+^ solution and thermal annealing. As a result, desired NWs with good morphology were obtained (Figure 8a–d). At 300 °C, the Fe_2_O_3_ NP-coated SnO_2_ NW sensor showed an almost six times higher response relative to the pristine sensor (Figure 8e), along with good selectivity to ethanol gas (Figure 8f). The presence of Fe_2_O_3_-SnO_2_ heterojunctions was attributed as the main reason for improved gas response, relative to the pristine sensor. Indeed, for the pristine sensor, the variations in EDL upon exposure to ethanol gas were not significant (Figure 9a). However, for the decorated sensor, the initial volume of the conduction channel was significantly decreased, owing to the presence of Fe_2_O_3_ NPs in the air. By subsequent exposure to ethanol, a remarkable change in the conduction channel inside the SnO_2_ NWs occurred (Figure 9b), resulting in a larger response than that from the pure sensor [66].

CuO is a promising candidate for H_2_S gas sensing due to direct reaction with H_2_S gas [67]. In this context, SnO_2_ NWs initially were synthesized at 900 °C for 10 min on a Au/Si substrate. Then, by sputtering, a Cu layer was coated onto the SnO_2_ NWs. Finally, branches of SnO_2_ NWs were synthesized at 300–700 °C for 30 min. For the NWs grown at 700 °C, CuO NPs were formed at their tips. The presence of CuO significantly enhanced the response to H_2_S. The primary reason for the enhanced selectivity to H_2_S is related to the reaction between CuO and H_2_S gases, as follows.
CuO + H_2_S = CuS + H_2_O(6)

Hence, CuO with semiconducting properties was transformed into CuS with high conductivity, which has much better conductivity than CuO. This significantly increased the resistance modulation [68]. In another study related to H_2_S gas sensing, SnO_2_ NWs were prepared at 900 °C for 1 h after the deposition of a Au layer on porous Si. In Figure 10a–f, the growth of SnO_2_ NWs on porous Si is shown. The results of H_2_S gas sensing showed an improved response of SnO_2_ NWs/porous Si relative to porous Si (Figure 10g–i). The heterojunctions between SnO_2_ and Si caused a flow of electrons from SnO_2_ to Si and vice versa. Hence, the hole conduction volume in the porous Si was considerably decreased. In H_2_S gas medium, due to the combination of electrons by holes, the hole resistance increased. Moreover, defects at the interfaces between the two materials were created due to lattice mismatch between them, leading to an enhanced response due to providing additional favorable adsorption sites [69].

Noble metal decoration via electronic and chemical sensitizations can boost the gas-sensing performance. In this regard, SnO_2_ NWs were synthesized by a similar method as described previously [62]. Then, SnO_2_ NWs were uniformly decorated with Pd NPs by radiolysis using ^60^Co γ-rays. Pd NPs in the range of 10–35 nm were uniformly deposited over the surface of SnO_2_ NWs. The Pd-decorated gas sensor showed a higher response to NO_2_ gas relative to the pristine sensor. The response was 19 compared to only 0.1 ppm NO_2_ gas and 66 to 1 ppm at 200 °C [70]. The good features of the Pd-decorated sensor are related to the catalytic activity of Pd NPs towards oxygen and NO_2_ gases, as well as the creation of Schottky contacts between Pd and SnO_2_ NWs. Modulation of the Schottky barriers in the NO_2_ atmosphere led to remarkable variations in the resistance and appearance of the sensing signal. Similarly, SnO_2_ NWs were grown on an electrode substrate. Then, Pd decoration was performed using UV-light irradiation (λ = 360 nm) of a PdCl_2_ solution in the presence of SnO_2_ NWs. By subsequent annealing at 500 °C, Pd NPs were dispersed on SnO_2_ NWs as isolated islands. Figure 11a–f show that SnO_2_ NWs with an approximate diameter of 50 nm and desired phases, along with Pd decoration, were successfully produced. The response to 10 ppm H_2_ gas at 300 °C was boosted by 141.4% after Pd decoration in addition to better selectivity to H_2_ gas. Pd NPs facilitated the dissociation of O_2_ and H_2_ gases. H_2_ decomposes to atomic H by a spillover effect, and it moves to the neighboring SnO_2_ surface by reaction with oxygen species, resulting in a significant decrease in sensor resistance (Figure 11g–h). Moreover, formation of Pd and SnO_2_ Schottky barriers led to the modulation of the sensor resistance in the H_2_ gas atmosphere due to a change in the height of barriers in the H_2_ gas atmosphere [71].

In another study, SnO_2_ NWs were prepared using a VLS mechanism on a Au-deposited Al_2_O_3_ substrate at 900 °C for 1 h. Pd embedding on SnO_2_ was used by applying a flame for 5 s at 1300 °C on PdCl_2_ containing SnO_2_ NWs. The sensor showed a higher response than the SnO_2_ NW sensor to NO_2_ gas, along with faster dynamics. The presence of oxygen vacancies, which are regarded as favorable sites for gas adsorption (Figure 12a), the formation of Schottky junctions between Pd and SnO_2-x_ (Figure 12b,c), and the spillover effect of Pd NPs (Figure 12d) all contributed to the response [72].

### 3.2. In_2_O_3_ NWs

In_2_O_3_ is an n-type metal oxide which is highly used for the sensing of different gases. Pristine and doped In_2_O_3_ NWs have also been widely synthesized by VLS growth [73,74,75]. By control of the temperature of precursors and source materials, different 1D In_2_O_3_ nanostructures can be synthesized [76,77]. Figure 13 schematically shows the dependency of the In_2_O_3_ morphology on the growth temperature [78,79]. Indeed, by careful control over the synthesis temperature, it is possible to obtain the desired morphology. Even catalyst-free growth of In_2_O_3_ NWs has been reported [80]. There have been studies involving the synthesis of In_2_O_3_ NWs by VLS growth for gas-sensing applications, which are reviewed below.

In_2_O_3_ powder was used for the growth of In_2_O_3_ NWs with the help of a Pd catalyst at 1500 °C for 1.5 h. The thinnest NWs were as narrow as 20 nm. The response of the sensor fabricated from these NWs was about 7 to 25 ppm of acetone at 400 °C [81]. However, both the growth temperature and sensing temperature were high. Moreover, no selectivity study was performed. In another study, In_2_O_3_ NWs were produced by the VLS mechanism. They had a length of 5 μm and diameters in the range of 10–100 nm. Two sensors were prepared from the single NW or NWs. The single NW sensor showed a response of 2.6 to 0.5 ppm at 200 °C, while the In_2_O_3_ multi-NW sensor revealed a response of 2.1 to 0.5 ppm NO_2_ gas [82]. However, it should be noted that making a sensing device with a single NW is challenging. To explore the effect of the electrode gap (10, 30, and 40 μm), In_2_O_3_ NWs were grown from metallic In powder on the sensor with different electrode gap sizes (Figure 14a–d). The growth was carried out at 700 °C for 0.5 h. The sensors were operated in a self-heating condition using a low voltage of 1.5 V. Among different sensors, the In_2_O_3_ NW sensor with an electrode gap of 40 µm exhibited the best response of 1.37 to 1000 ppm along with response and recovery times of less than 12 and 35 s, respectively [83]. Moreover, the consumed power increased with a decrease in the electrode gap size. Indeed, with a small gap (10 and 30 µm), the density of the NWs between the gap was highly dense, leading to high power consumption. Despite this, in the large gap (40 µm) electrode, the density of the NWs was low, resulting in low power consumption.

In another study, In_2_O_3_ NWs were synthesized at 800 °C for 1 h, and the effect of the flow rate of the carrier gas on the morphology of NWs was explored. A low flow rate (50 mL min^−1^) of the carrier gas resulted in an insufficient supply of gas to the substrate; hence, In_2_O_3_ NPs were formed. By increasing the flow rate to 100 mL min^−1^, the supply of atoms was increased to form the 1D In_2_O_3_ NWs. Hence, due to the higher reactive stick possibility of gas on the catalyst droplet and precipitation at the catalyst–NW interface, high-aspect-ratio NWs with lengths of 1–5 μm and diameters of 100 nm were synthesized. However, the increase in the flow rate to 100 mL min^−1^ led to the formation of microrods with a low aspect ratio. Using In_2_O_3_ NWs, a gas sensor was fabricated which showed a low response of 1.2 to 1000 ppm H_2_ gas at 250 °C along with a sensor t_res_ of 48 s and a t_rec_ of 176 s [84]. However, the response was very low for practical applications. In another interesting study, Au-catalyst-loaded nickel foams were initially synthesized by Au sputtering for 5, 10, 20, and 30 s. Then, using In and Sn powders (mass ratio of 10:1), Sn-doped indium In_2_O_3_ NWs were fabricated at 835 °C; for 0.5 h. At 125 °C, the sensor prepared with 10 nm-thick Au catalyst showed the highest sensing response of 160 to 100 ppm ethylene glycol along with fast response and recovery times of 8.7 s and 19.3 s, respectively. For a Au sputtering time of 5 s, the density of NWs on NiO was small. Moreover, for sputtering times of 20 and 30 s, NWs completely covered the NiO surface and significantly decreased the number of p-NiO/n-Sn-doped In_2_O_3_ heterojunctions, causing a decrease in the sensing response. Under the optimal sputtering time (10 s), sufficient amounts of p–n junctions were formed, leading to improved gas response [85]. Furthermore, the Ni^3+^ ions had a catalytic effect, and the highest amount of adsorbed oxygen and oxygen vacancies in the optimal sensor contributed to the sensing signal. To synthesize Zn-doped In_2_O_3_, ZnO and In_2_O_3_ powders with graphite powder were heated at 900 °C for 30 min to grow Zn-doped in_2_O_3_ NWs on a Au-coated (9 nm) alumina substrate. The optimized molar fraction of Zn was 3% since higher amounts of Zn resulted in the formation of a ZnO phase. Due to the different sizes of Au catalysts, NWs with different diameters (70–150 nm) were obtained. Interestingly, at high temperatures (300 °C and above), the sensor showed an unusual n-to-p transition. Zn-doping can significantly increase the oxygen vacancies in In_2_O_3_ and surface defects. Higher defects led to more adsorption of oxygen species from ambient air. At high temperatures, adsorbed oxygen ions diffused into the bulk regions and filled the oxygen vacancies, leading to a decrease in electrons. Furthermore, capturing of electrons by NO_2_ gas led to a further decrease in the electrons. Hence, a down shift of the Fermi level (E_F_) in NWs occurred. In addition, by Zn-doping, Zn^2+^ replaced In^3+^ and contributed to the downward shift of the E_F_. The down shift of the E_F_ caused the transfer of electrons in the opposite path, resulting in an opposite sensor response at high temperatures. Zn-doped In_2_O_3_ NWs showed a 30 times higher response to ethanol relative to the pristine sensor, due to the high oxygen vacancies and higher defects in Zn-doped In_2_O_3_ NWs [86].

Using In_2_O_3_ powder, In_2_O_3_ NWs were grown at 1500 °C for 1.5 h. The growth was performed on bare alumina and alumina seeded by metallic “In” NPs. For the former case, micrometer-sized nanostructures were formed; in the second case, connected NWs were formed. The NWs gas sensor had higher responses to acetone and NO_2_ gases relative to a thick wire gas sensor at 400 °C owing to the higher surface area of NWs, which provided a lot of adsorption sites for the incoming gas [87]. Growth of In_2_O_3_ nanostructures using In based on the VLS mechanism on a Au-coated SiO_2_ substrate was carried out at 700–900 °C for 1 h. At 700 °C, uniform In_2_O_3_ nanostructures were grown. At 800 °C, the diameter and length of the NWs were 70–80 nm and several micrometers, respectively. The product displayed a high density of NWs. At 900 °C, In_2_O_3_ nanoneedles with diameters of 150–200 nm and lengths about 4–5 μm were formed. H_2_ gas-sensing results showed that the In_2_O_3_ NWs had a higher response relative to In_2_O_3_ nanoneedles due to the high surface area of In_2_O_3_ NWs relative to nanoneedles. However, the sensor temperature was high at 400 °C [88]. In_2_O_3_ NWs were prepared at 800 °C for 1 h on a Au-coated (3 nm) Si substrate. Then, Pt was sputtered onto In_2_O_3_ NWs, and by subsequent annealing, Pt islands were formed on In_2_O_3_ NWs. At 50 °C, decorated NWs displayed a response of 4 to 400 ppm O_2_ gas, while the pristine sensor showed no response to this gas. Indeed, oxygen species were adsorbed and disassociated on Pt NPs and then migrated onto the In_2_O_3_ NWs, where they were adsorbed, leading to a high response to oxygen gas [89]. Using a mixture of In_2_S_3_ and In_2_Se_3_ powders (10:1 volume ratio), In_2_O_3_ NWs were synthesized by the VLS mechanism on a Au-coated (3 nm) Si wafer at 850 °C for 1 h. Second, a solution of Ti^4+^ ions was spin-coated onto an In_2_O_3_ NW template, and after being prepared at 500 °C for 2 h, TiO_2_ NPs (10–30 nm) were dispersed on the surface of NWs. As shown in Figure 15a–d, the synthesized NWs had high aspect ratios (diameters = 50 to 100 nm).

Figure 15e displays the acetone gas-sensing performance of both gas sensors at 250 °C. Clearly, the TiO_2_-decorated In_2_O_3_ NW gas sensor displayed a higher response to acetone relative to the bare gas sensor. Moreover, both gas sensors showed a high stability over a period of 30 days (Figure 15f). SnO_2_-SnO_2_ homojunctions in both gas sensors contributed to the sensing signal. However, in the decorated gas sensor, the formation of TiO_2_-In_2_O_3_ heterojunctions provided additional sources of the resistance modulation, causing a higher sensing performance relative to the pure gas sensor. Furthermore, the surface areas of pristine and TiO_2_-decorated NWs were 4.25 and 6.72 m^2^/g, respectively, leading to more adsorption sites on the decorated gas sensor, contributing to the sensor signal [90].

In and Mg mixed powders (weight ratio of 1:1) were taken for the growth of In_2_O_3_ NWs, in which Mg lowered the melting point of the In-Mg alloy. The NWs were prepared at 800 °C for 1 h. Then, Pd was sputtered on In_2_O_3_ NWs, and subsequent heat treatment was carried out at 700 °C, resulting in the dispersion of Pd NPs on the surface of NWs. The decorated sensor showed a higher response to NO_2_ gas relative to the bare sensor. Enhanced sensing features were related to the catalytic effect of Pd on O_2_ and NO_2_ gases as well as the generation of Schottky junctions between Pd and In_2_O_3_ in intimate contact [91]. In another study, In_2_O_3_ NWs were grown at 800–1000 °C on a Au (3 nm)-coated SiO_2_/Si substrate using metallic In powders. At 800 and 850 °C, NWs were not formed. As the growth temperature enhanced, the length increased while the diameter of the NWs decreased. Indeed, NWs with a higher aspect ratio were prepared at high temperatures due to the larger indium vapor pressure. At 300 °C, the responses to 100 ppm ethanol gas for the samples grown at 900, 950, and 1000 °C were 1.84, 3.78, and 13.97, respectively. The good response of the optimal gas sensor was related to the narrower diameter of NWs grown at 1000 °C, leading to a larger surface area. Moreover, for narrower NWs, most of the NWs were depleted of electrons, and by exposure to ethanol gas, significant variation in the width of the EDL leads to a higher response. In addition, the optimal sensor showed good selectivity to ethanol in the presence of methanol and acetone gases [92].

### 3.3. ZnO NWs

ZnO is a highly popular semiconducting metal oxide for gas-sensing usages, thanks to its high mobility of carriers, high stability, ease of synthesis, and low price. Using a mixed Zn and Sn powder, ZnO NWs were grown on glass substrates with interdigitated ITO electrodes where Sn was used as the catalyst for the growth of ZnO NWs at 900 °C. After the growth of ZnO NWs, Fe_2_O_3_ NPs were attached to them using a chemical process. A α-Fe_2_O_3_/ZnO NW gas sensor showed a higher response to CO gas relative to both pristine Fe_2_O_3_ and ZnO gas sensors. At 300 °C, it revealed a response of 18.8 to 100 ppm CO gas. The formation of heterojunctions between α-Fe_2_O_3_ and ZnO and the high surface area of ZnO NWs caused an enhanced gas response relative to other gas sensors [93]. Branching NWs offer a high surface area along with many heterojunctions, which are highly demanded for sensing studies. In a relevant study, core SnO_2_ NWs were synthesized at 1050 °C for 2 h on a Si substrate. Then, different thicknesses (1, 3, 10, and 30 nm) of Au were deposited over the SnO_2_ NWs, and ZnO branches on SnO_2_ were grown at 500 °C for 1 h [94]. Based on the SEM images presented in Figure 16a–d, when using a thick Au layer (10 and 30 nm), no branched structures were formed. Moreover, the diameters of the branches grown using Au with thicknesses of 1 and 3 nm were 30 and 34.5 nm, respectively. The thin (1, 3 nm) Au layers resulted in small and disconnected particles upon heating, leading to the growth of thin ZnO branches (Figure 16e). However, thick (10 and 30 nm) Au layers resulted in the formation of a connected and rough Ag layer upon heating. Hence, nuclei interfere during the initial growth, forming thick cluster-like ZnO (Figure 16f). A sensor was fabricated from the branched NWs, and at 250 °C it showed a response of 6 to 100 ppm NO_2_ gas. The formation of ZnO-ZnO homojunctions along with the formation of SnO_2_-ZnO heterojunctions was attributed to the sensing mechanism. The change in the barrier height of junctions in the NO_2_ gas atmosphere significantly modulated the sensor resistance [94].

In another study, the growth of ZnO NWs using Zn powders was performed in the presence of Au-coated Si substrates at 900 to 1050 °C for 1 h to explore the impact of temperature. Figure 17a,c,g,e show SEM micrographs of ZnO NWs synthesized at 900, 950, 1000, and 1050 °C, respectively. According to Figure 17b,d,f,h, the diameters of NWs grown at the aforementioned temperatures were 200–550, 90–190, 50–150, and 30–90 nm, respectively.

Accordingly, with increasing growth temperature, NWs with smaller diameters were synthesized thanks to the faster diffusion of atoms along the NWs, instead of remaining on the stems of the NWs, resulting in the creation of thinner NWs. Figure 18a–c shows a typical TEM and TEM-EDS analysis of a single NW, demonstrating that the growth was performed using VLS analysis, where Au NPs at the tip of a ZnO NW were detected. The sensor fabricated from the sample with the smallest diameter was used for oxygen-sensing studies at 25 °C. It showed a sensitivity of 100% to 100 ppm oxygen gas. However, no selectivity study was performed [95].

In another study, ZnO NWs were directly grown at 950 °C for 1 h using the VLS method on an electrode substrate. Tri-layered Ti/Pt and Au (5 nm) electrodes were applied on a SiO_2_/Si substrate. For Pt decoration using sputtering, the thickness of the Pt (5 and 10 nm) and heat-treating temperature (500–750 °C) were changed, whereas for Pd decoration, the UV irradiation time (1, 5, 10, and 20 s) was varied. Figure 19a,b demonstrate the different steps for the synthesis and sensing fabrication of Pd or Pt-decorated gas sensors. The particle size of Pt on ZnO NWs increased with increasing annealing temperature. For samples calcined at 600 and 650 °C, the sizes were 18 and 28 nm, respectively, for the 10 nm Pt layer and 20 and 30 nm, respectively, for the 5 nm Pt layer. Moreover, for Pd NPs, under 5 s of irradiation, the largest particle size of 24 nm was resulted. At 25 °C, under the self-heating mode with 20 V of applied voltage, the highest response of Pt-decorated (initial thickness = 5 nm, annealed at 600 °C) ZnO NWs to 50 ppm toluene was 2.86. In addition, the highest response of Pd-decorated (UV irradiation time, 5 s) ZnO NWs to 50 ppm benzene was 2.20. Therefore, it was possible to realize selective benzene and toluene gas sensors using the decoration of Pd and Pt on ZnO NWs. Chemical sensitizations of Pd and Pt NPs as well as the formation of Schottky junctions between Pd and ZnO and Pt and ZnO resulted in enhanced gas performance [96].

Using the VLS growth mechanism, ZnO NWs were fabricated directly on alumina with Au electrodes at 900 °C for 20 min (Figure 20a). Through the cation exchange and by evaporation of NiCl_2_ at 700 °C, ZnO NWs were transformed into NiO NWs (Figure 20b). Then, ZnO NWs were grown on NiO NWs using Zn powder at 500 °C for 10 min. Finally, Ni-doped ZnO-branched NiO NWs were formed (Figure 20c). Figure 21a–d show different characterizations of the synthesized materials, demonstrating the formation of materials with favorable branched morphology and composition.

The fabricated sensor displayed a response of 42.44 to 5 ppm p-xylene at 400 °C, which is better than pristine ZnO and NiO NWs along with good selectivity (Figure 22). The improved sensitivity of Ni-doped branched ZnO NWs was attributed to the presence of a lot of Schottky barriers formed between the branches, and the selectivity to xylene was related to the catalytic performance of the Ni dopant [97]. Similarly, Co-doped ZnO-branched ZnO NWs were prepared using the same procedure described previously [97] using CoCl_2_ powders instead of NiCl_2_ powders. The response to 5 ppm p-xylene at 400 °C was ~19.60. Again, the improved sensitivity to p-xylene was attributed to the catalytic effect of Co and the formation of plenty of Schottky barriers between branches and stems [98].

In another research work, ZnO NWs were directly prepared at 950 °C for 1 h on a SiO_2_/Si substrate with electrodes consisting of Ti/Pt and Au (3 nm) (Figure 23a,b). Then, γ-ray radiolysis using ^60^Co was employed for Pd decoration. A dose rate of 10 kGy h^−1^ for 3 h and subsequent annealing at 500 °C for 1 h in air resulted in the formation of Pd with an average diameter of 20 nm on ZnO NWs (Figure 23c,d).

At 24 °C, it displayed a response of 1.02 to 100 ppb CO gas, while the pristine sensor did not show any response. Generally, Pd is oxidized to PdO in air, and in the presence of CO gas it is reduced to Pd. Hence, the electron flow between the Pd/PdO phase and the ZnO NWs affected the sensor response. The catalytic activity of Pd to Co also affected the sensing performance [99].

ZnO NWs were synthesized on Ti-Pt electrode SiO_2_/Si substrates via the VLS growth mechanism. Subsequently, Au layers (3, 5, 10, 20, and 30 nm) were sputter-deposited onto the ZnO NWs followed by annealing. The size of the Au NPs increased with an increase in the initial Au thickness from 3 to 30 nm. The sensor with a 10 nm-thick Au displayed the largest response to CO gas at 7 V in the self-heating mode. The catalytic effect of Au towards CO gas and the formation of Schottky barriers between Au and ZnO led to an enhanced gas sensor [100]. Networked ZnO NW sensors were grown at 950 °C for 1 h via growth by the VLS mechanism. They were directly grown on SiO_2_-/Si substrates having tri-layered electrodes consisting of Au (3 nm)/Pt (100 nm)/Ni (50 nm) with different electrode gaps of 10, 20, 50, and 60 μm. In this way, it was possible to control the number of junctions between the as-grown NWs. For the thinnest spacing of 10 μm, networked NW junctions formed more densely, and by increasing the electrode gap, less dense junctions were formed. Moreover, for the electrodes with gaps of 60 μm, no junctions were formed. Therefore, the narrower the gap between electrodes, the denser the junctions between ZnO NWs. Based on the sensing results, at 350 °C, the sensor with a smaller gap exhibited a higher response to NO_2_ gas thanks to the creation of more ZnO-ZnO homojunctions and, consequently, more modulation of the sensing resistance [101].

Because of their novel structure that includes the head group, backbone, and end group, self-assembled monolayers (SAMs) have different functionalities, leading to the possibility of their use for the detection of different gases [102]. In this regard, ZnO NWs with diameters ranging from 10 to 20 nm were grown using a Au catalyst VLS mechanism with ZnO powders at 1200 °C for 12 min and then functionalized with 3-aminopropyl trimethoxysilane (APTMS) and 3-glycidoxypropyltrimethoxysilane (GLYMO) SAMs. The 10 × 10^−3^ m APTMS-functionalized ZnO NWs sensor displayed an improved response to acetone at 300 °C along with acceptable selectivity. The sensing enhancement was related to the significant modulation of the EDL width and the reaction between the amino groups (–NH_2_) of APTMS SAMs and acetone molecules. Indeed, the reaction between the acetone electrophilic “C” atom and nucleophilic (-NH_2_)^−^ takes place, resulting in the formation of imine ((H_3_C)_2_-C=N) and H_2_O molecules, eventually modulating the resistance [103].

### 3.4. Other Metal Oxide NWs

In addition to SnO_2_, In_2_O_3_, and ZnO NWs, some other semiconducting metal oxides have also been synthesized by VLS growth for gas-sensing usages. We will review and discuss them. Moumen et al. [104] prepared crystalline Bi_2_O_3_ NWs via the VLS mechanism using three catalysts. Pt, Cu, and Au thin layers with thicknesses of less than 3 nm were sputter-deposited onto Al_2_O_3_ substrates. Using Bi_2_O_3_ powder, the growth of nanostructures was performed at 1000 °C, and the substrate temperature was set to 550 °C. Figure 24a–c show SEM images of Bi_2_O_3_ nanostructures grown using different catalysts, namely, Au, Pt, and Cu, respectively. NWs with ~100 nm diameters and 6 μm lengths were grown using a Au catalyst (Figure 24d,e), while micropillars with irregular shapes were grown using a Pt catalyst. The use of a Cu catalyst led to the formation of few microrods.

Gas-sensing measurements using Bi_2_O_3_ NWs demonstrated that the sensor was more sensitive to ethanol gas (Figure 24f). Homojunctions between Bi_2_O_3_-Bi_2_O_3_ NWs acted as a resistance modulation source during exposure to ethanol gas. Furthermore, the presence of Au NPs at the top of NWs enhanced the sensing performances due to their catalytic effect towards the dissociation of oxygen and ethanol gases as well as the formation of potential barriers with Bi_2_O_3_ NWs. Moreover, WO_3_ NWs are a good candidate to be grown by the VLS mechanism. In this regard, using WO_3_ powder, WO_3_ NWs were grown at 1100 °C over Pt or Au catalysts with the same thicknesses (3 nm) on alumina substrates [105]. Moreover, the temperature of substrate was varied between 500 and 700 °C, and the deposition times were set to 15 and 20 min. In the case of the Pt catalyst, no growth of NWs occurred at 525 and 580 °C, while at 700 °C NRs were grown. In the case of the Au catalyst, NWs were grown at 525 °C as the optimal substrate temperature, and they had small diameters (10–30 nm). Furthermore, the WO_3_ NWs prepared over a longer deposition time were denser, with their length in the micrometer range. However, NWs prepared for 15 min revealed lengths and diameters in the nanometer range. The WO_3_ NW sensor was prepared using optimized WO_3_ grown using a Au catalyst for 15 min. It displayed a response of 103 to 10 ppm of H_2_S at 400 °C and a high response of 170 to 300 ppb ozone at 200 °C. In particular, the high response to ozone was related to the dipole moment of ozone and its central atom, which has a positive charge, and the other side atoms, which have a negative charge. Upon interaction of the central atom with the WO_3_ surface, it accepted electrons from WO_3_, and the other two atoms formed O_2_ gas. Thanks to the high electron affinity of ozone, the amount of adsorbed oxygen species on the surface of the sensor increased, causing an increase in the EDL thickness and resistance in the presence of ozone. Kaur et al. [106,107] synthesized NiO NWs on an alumina substrate by the VLS mechanism using Au, Pt, and Pd catalysts for gas-sensing studies. The growth was performed at 1400 and 1450 °C, and the temperature of the substrate was varied between 900 and 1000 °C. For both Pt and Pd catalysts, NWs along with some other nanostructures were formed, while the use of a Au catalyst resulted in the formation of only NiO NWs. Under optimal conditions with Au catalysts, i.e., growth at 1400 °C and a substrate temperature of 930 °C, NiO NWs with small diameters (16–50 nm) were fabricated. The NiO sensor showed its best performance to acetone and ethanol gases at 500 °C and H_2_ and CO gases at 300 °C. Hence, the working temperature can tune the selectivity to different gases. However, the working temperatures were relatively high, especially for acetone and ethanol detection, resulting in high power consumption in real applications. In another work, NiO NWs were synthesized using NiO powder at 1400 °C for 15 min on Au/alumina substrates. Then, ZnO NWs (with a size of 9 nm) were grown on NiO NWs without any catalyst at 1200 °C. At 400 °C, the branched sensor showed almost twice the response for both acetone and ethanol gases, relative to pristine NiO NWs [108]. *β*-Ga_2_O_3_ nanostructures have also been prepared by the VLS process [109]. In a study, single-crystalline *β*-Ga_2_O_3_ NWs on silicon (100) substrates were fabricated using Ga_2_O_3_ powder at 975, 1000, 1025, and 1050 °C for 30 min. Then, Au decoration was performed on the synthesized NWs. The diameter of the NWs was increased from 122 to 133 nm with growth temperature. A gas sensor was realized from the sample grown at 1025 °C. The CO gas-sensing features of Au decorated on multiple *β*-Ga_2_O_3_ NWs with Au-decorated single *β*-Ga_2_O_3_ NW and single pristine *β*-Ga_2_O_3_ NW were compared at RT. It was found that the sensor fabricated from a single Au-decorated *β*-Ga_2_O_3_ NW showed an enhanced response to CO gas with faster dynamics [110].

The VLS growth mechanism using evaporation of Fe sources has been less studied. In fact, due to the high melting points of metallic Fe and its oxides, Fe_2_O_3_ NWs, which are thermodynamically the most stable type of iron oxide, are often produced by thermal oxidation routes using oxidation of Fe foil [111,112,113,114]. In a rare report about the VLS growth mechanism of iron oxide NWs in 2006, PLD-assisted VLS growth resulted in iron oxides with different morphologies such as NRs, NWs, and nanobelts. Fe_3_O_4_ in pellet form was used as the source material. Then, it was evaporated by laser beams. By PLD, it is possible to maintain the stoichiometry of the source material due to the high heating rate. Furthermore, it needs a lower substrate temperature relative to other evaporation techniques [115]. In this way, Fe_2_O_3_ NWs were formed. However, the synthesized NWs were not used for sensing studies. Hence, it seems that VLS-grown Fe_2_O_3_ NWs for gas sensing require further studies.

Table 1 provides the gas-sensing performance of different gas-sensing materials synthesized by the VLS growth mechanism. It can be seen that metal oxides synthesized by this method can be successfully used for the detection of various gases at different temperatures.

Metal oxide NW gas sensors used as resistive gas sensors have advantages such as high response, high stability, fast response and recovery times, low price, and ease of fabrication and use, relative to other types of gas sensors [33]. Moreover, in comparison to other 1D morphologies such as NF gas sensors [116], NWs generally have better crystallinity, and they can be easily branched so that their surface area can be further increased [117]. However, the gas sensors with NW morphology are still in a non-commercial state, mainly due to non-uniform growth and poor control over density and the distribution of as-synthesized NWs [118].

## 4. Conclusions

In this review paper, we discussed the VLS growth mechanism of semiconducting metal oxides for gas-sensing applications. The synthesis of SnO_2_, In_2_O_3_, and ZnO NWs via the VLS growth mechanism for gas-sensing applications is the most reported relative to other gas sensors such as Bi_2_O_3_, Ga_2_O_3_, and WO_3_. This mechanism is mostly suitable for metals or metal oxides with melting points lower than ~1400 °C. Hence, no studies have been performed on the synthesis of Fe_2_O_3_ NWs grown by the VLS mechanism for gas-sensing studies yet. By controlling the growth temperature, catalyst type, and growth time, the formation of NWs with desired diameters and aspect ratios is possible. Different metal catalysts are employed for the growth of NWs. However, Au is the most commonly used catalyst due to its inertness and ease of coating on substrates as well as the formation of a low-temperature eutectic Si substrate, which is the most common type of substrate. VLS-grown metal oxides can be successfully used for the detection of various gases, depending on the synthesized materials, sensing temperature, and nature of the target gas. Generally, branched NWs as well as hierarchical NWs result in the realization of gas sensors with higher performance relative to NWs due to having a larger surface area along with the presence of more homojunctions. In particular, for branched NW when the core and branch materials are synthesized from different materials, with the formation of heterojunction barriers, the core and branch NWs can significantly affect the gas-sensing properties. Moreover, decoration of noble metals on the surface of NWs can improve the sensing properties due to the catalytic effect of noble metal NPs as well as the formation of Schottky barriers in the interfaces between them. Doping is another strategy which can improve the sensing properties of metal oxide NWs. Generally, it leads to the formation of structural defects such as oxygen vacancies, which are considered good sites for gas adsorption. In future studies, the effect of high energy irradiation, bimetallic decoration, and hybrid formation with conducting polymers and graphene materials can be explored for gas-sensing applications.

## Figures and Tables

**Figure 1 materials-16-06233-f001:**
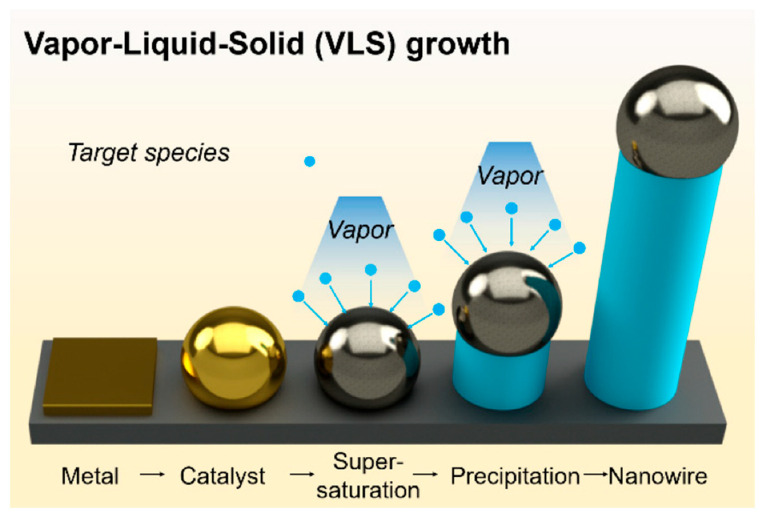
VLS mechanism for the growth of NWs [35].

**Figure 2 materials-16-06233-f002:**
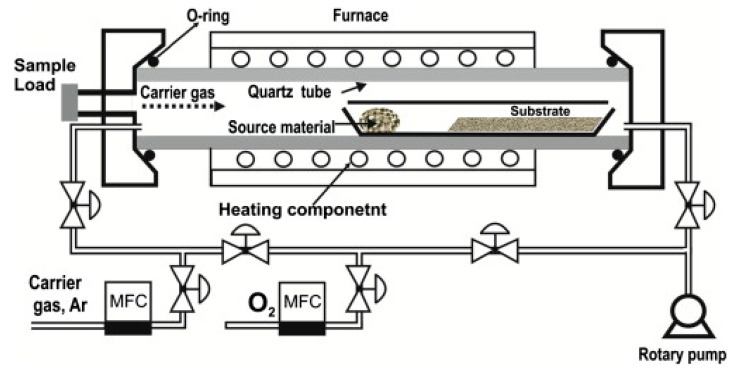
Schematic of the synthesis setup for the growth of metal oxide NWs [37] (Used with permission from Elsevier^®^).

**Figure 3 materials-16-06233-f003:**
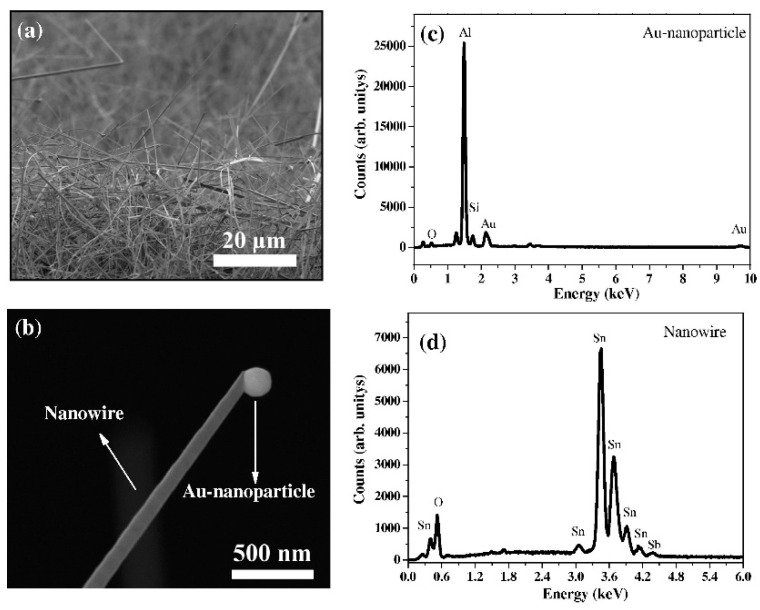
SEM micrographs of (**a**) Sb-doped SnO_2_ NWs and (**b**) a single Sb-doped SnO_2_ NW. EDS spectra of (**c**) Au NPs and (**d**) Sb-doped SnO_2_ NWs [54] (Used with permission from Elsevier^®^).

**Figure 4 materials-16-06233-f004:**
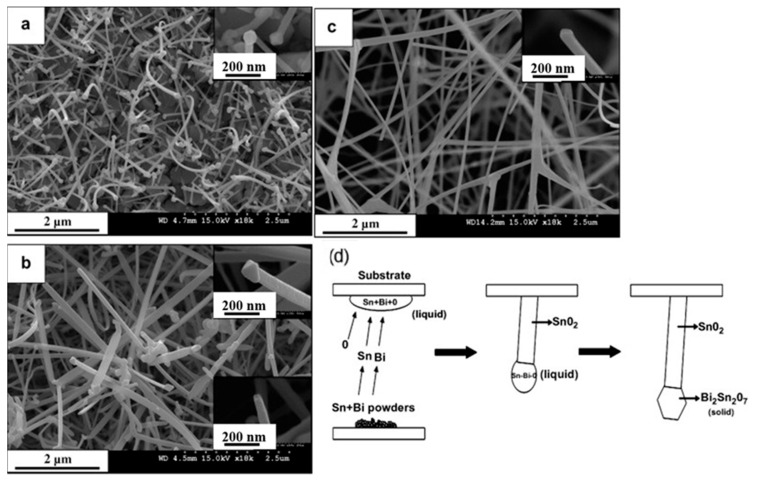
SEM images of SnO_2_ NW–Bi_2_Sn_2_O_7_ NP composites grown at (**a**) 700, (**b**) 800, and (**c**) 900 °C. The insets show SEM images of the tips of NWs. (**d**) Schematic of the growth mechanism [56] (Used with permission from Elsevier^®^).

**Figure 5 materials-16-06233-f005:**
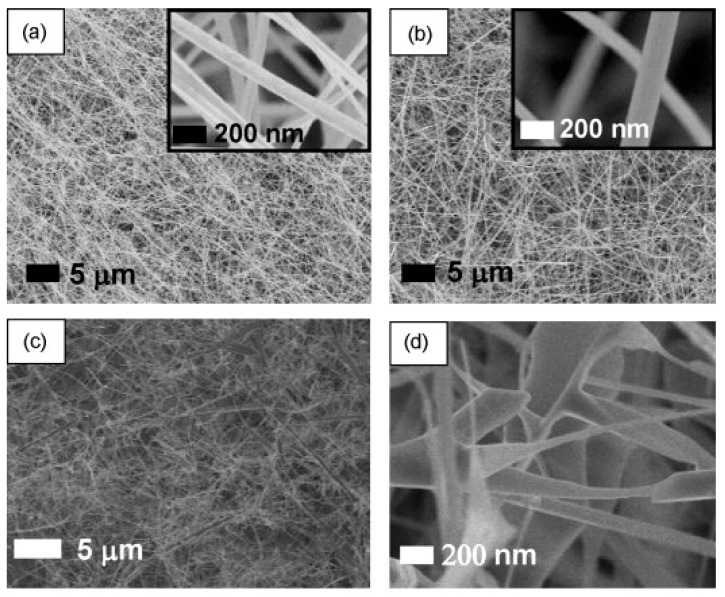
FE-SEM micrographs of SnO_2_ NWs grown at (**a**) 980 and (**b**) 800 °C and (**c**,**d**) hierarchical SnO_2_ nanostructures [59]. The insets show higher magnification images (Used with permission from Elsevier^®^).

**Figure 6 materials-16-06233-f006:**
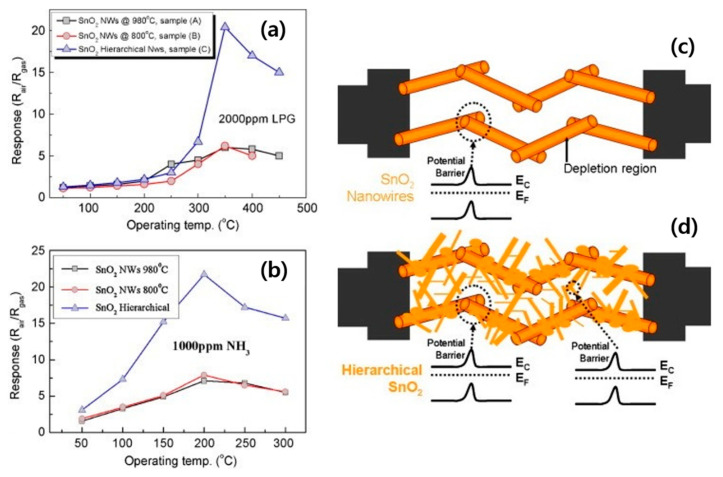
Response versus temperature for different gas sensors to (**a**) LPG and (**b**) NH_3_ gases [59]. The sensing mechanism of (**c**) SnO_2_ NWs and (**d**) hierarchical SnO_2_ [59] (Used with permission from Elsevier^®^).

**Figure 7 materials-16-06233-f007:**
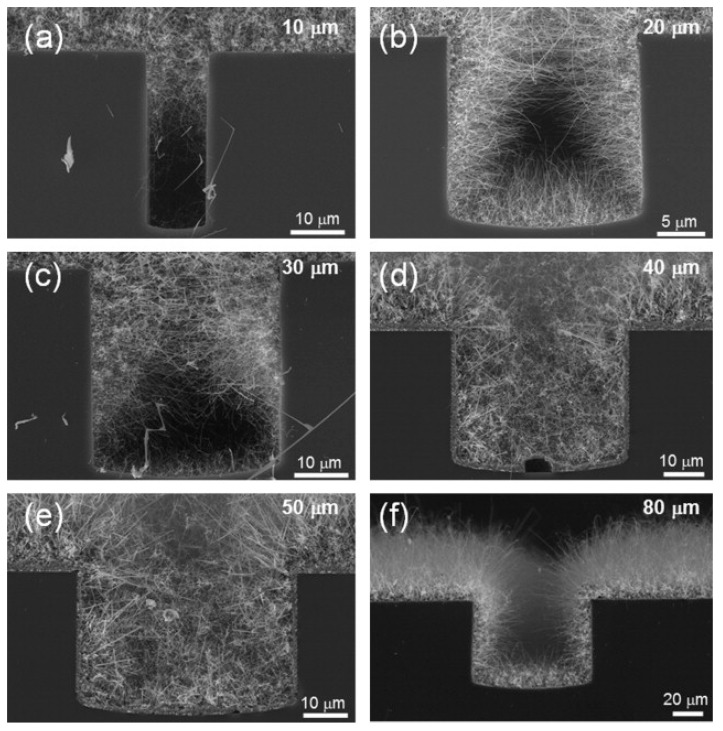
FE-SEM images of SnO_2_ NWs grown on substrates with different trench widths of (**a**) 10, (**b**) 20, (**c**) 30, (**d**) 40, (**e**) 50, and (**f**) 80 μm [60] (Used with permission from Elsevier^®^).

**Figure 8 materials-16-06233-f008:**
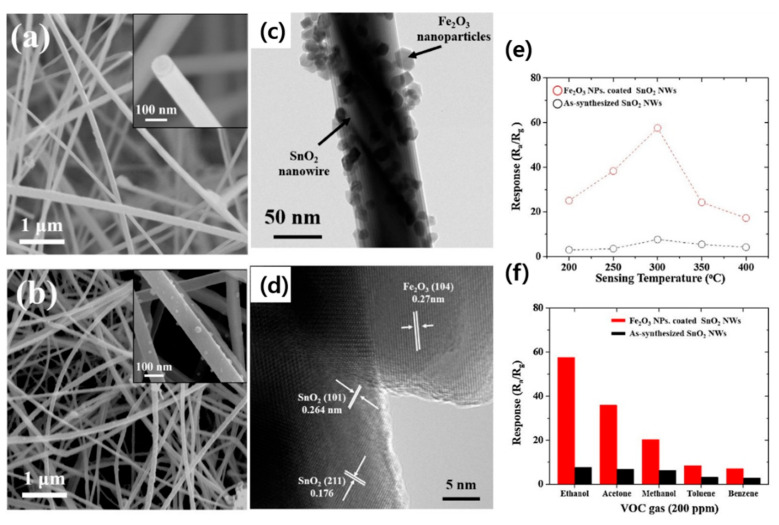
SEM micrographs of (**a**) pure SnO_2_ NWs and (**b**) Fe_2_O_3_ NP-coated SnO_2_ NWs along with (**c**) TEM and (**d**) HR-TEM images [66]. The insets are larger magnification images. (**e**) Response to 200 ppm ethanol gas versus temperature and (**f**) selectivity patterns of gas sensors [67] (Used with permission from Elsevier^®^).

**Figure 9 materials-16-06233-f009:**
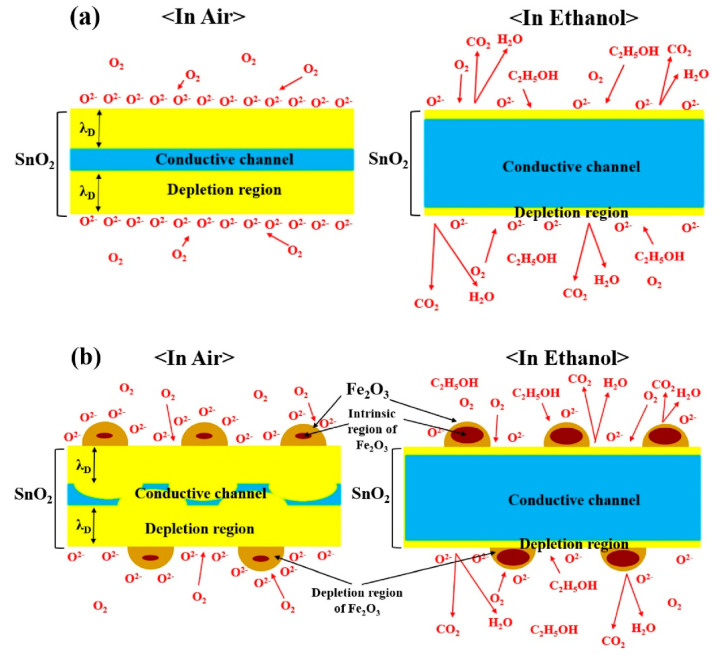
Schematic of the ethanol sensing of (**a**) pure and (**b**) Fe_2_O_3_ NP-coated SnO_2_ NW sensors [66] (Used with permission from Elsevier^®^).

**Figure 10 materials-16-06233-f010:**
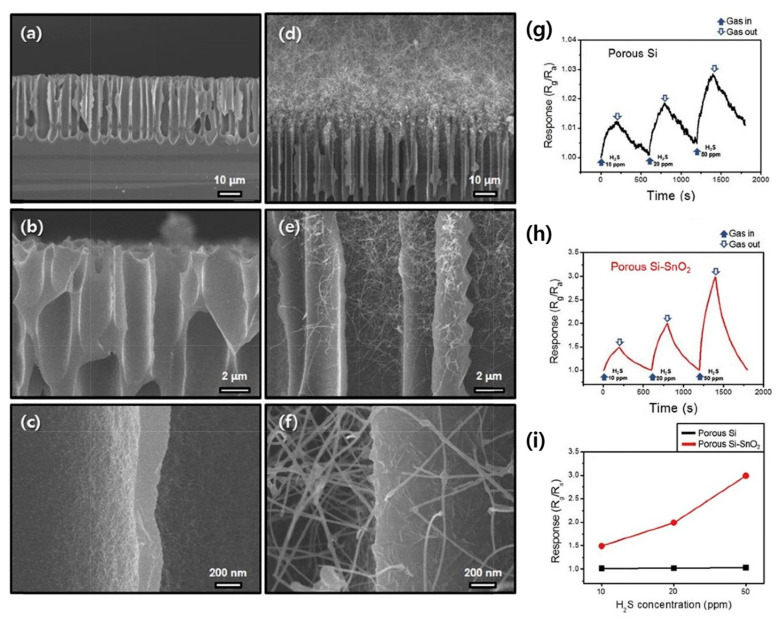
SEM micrographs of (**a**–**c**) porous Si and (**d**–**f**) porous Si-SnO_2_ NWs [70]. Dynamic response graphs to H_2_S gas at 100 °C for (**g**) porous Si and (**h**) porous Si-SnO_2_ NW sensors. (**i**) Calibration curves of both gas sensors [69] (Used with permission from Elsevier^®^).

**Figure 11 materials-16-06233-f011:**
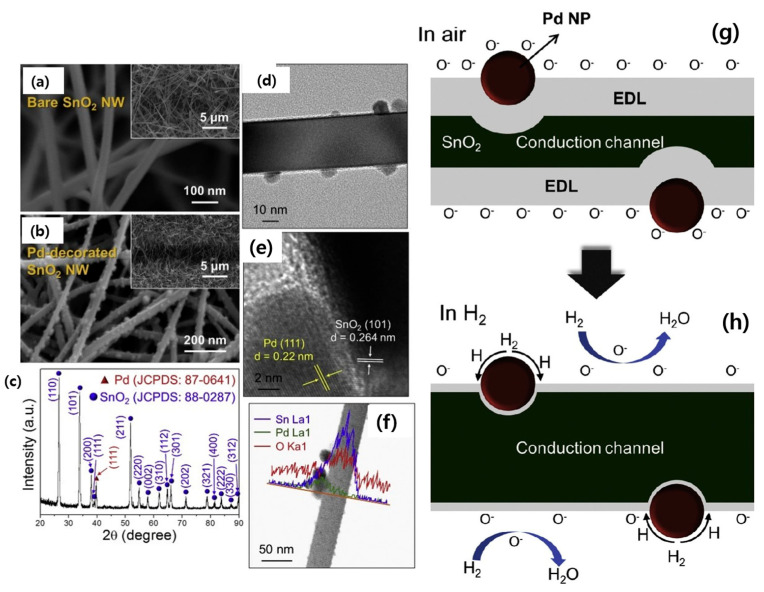
FE-SEM images of (**a**) SnO_2_ NWs and (**b**) Pd-SnO_2_ NWs. The insets show lower magnification images. Characterizations of Pd-decorated SnO_2_ NWs: (**c**) XRD pattern, (**d**) TEM image, (**e**) HRTEM image, and (**f**) EDS line mapping. (**g**,**h**) Sensing mechanism of Pd-decorated SnO_2_ NWs [71] (Used with permission from Elsevier^®^).

**Figure 12 materials-16-06233-f012:**
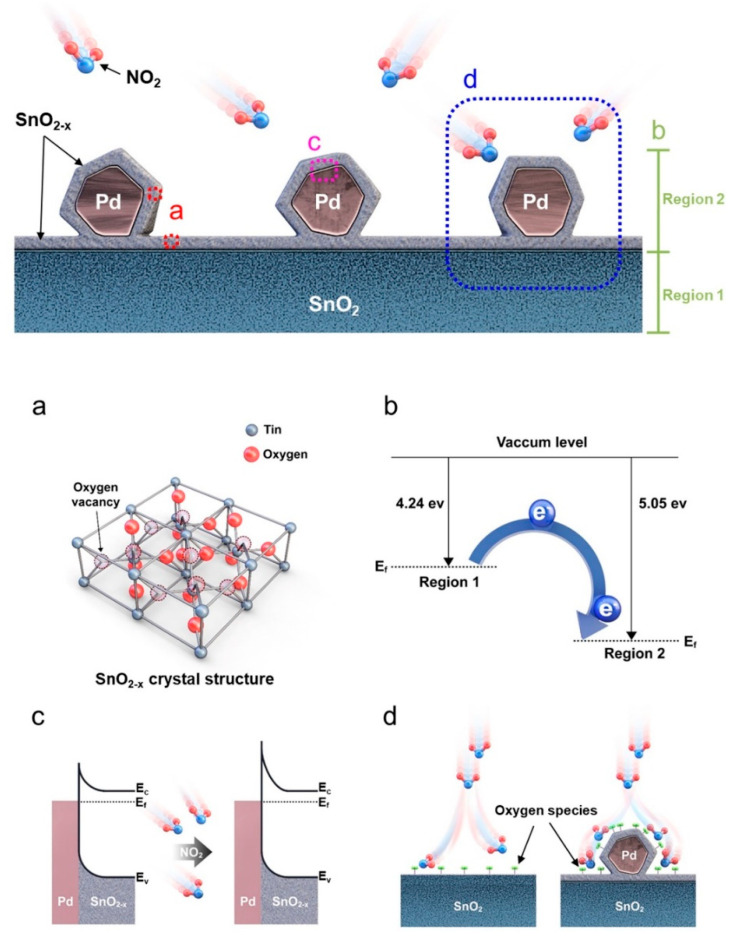
(**a**) SnO_2-x_ with oxygen vacancies, (**b**) energy levels, (**c**) formation of Schottky barriers, and (**d**) spillover effect of the catalyst [72] (Used with permission from Elsevier^®^).

**Figure 13 materials-16-06233-f013:**
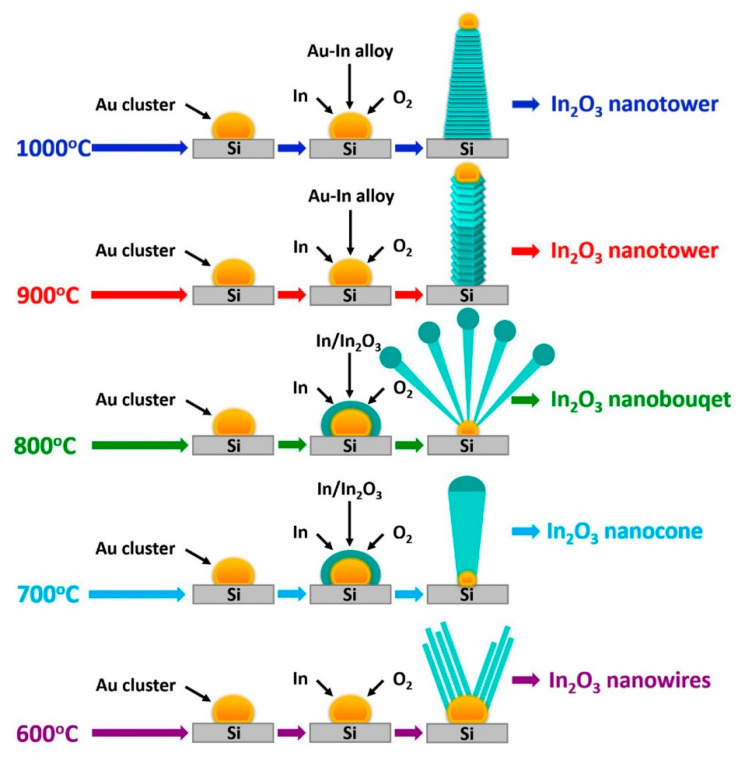
One-dimensional In_2_O_3_ nanostructures synthesized by the VLS growth method [79] (Used with permission from Elsevier^®^).

**Figure 14 materials-16-06233-f014:**
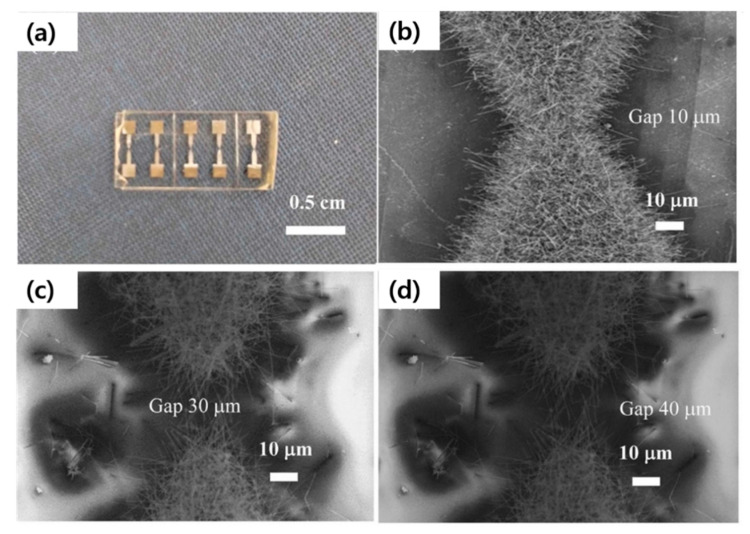
(**a**) Sensors with various electrode gaps. SEM images of In_2_O_3_ NWs grown with different electrode gaps of (**b**) 10, (**c**) 20, and (**d**) 40 μm [83] (Used with permission from Elsevier^®^).

**Figure 15 materials-16-06233-f015:**
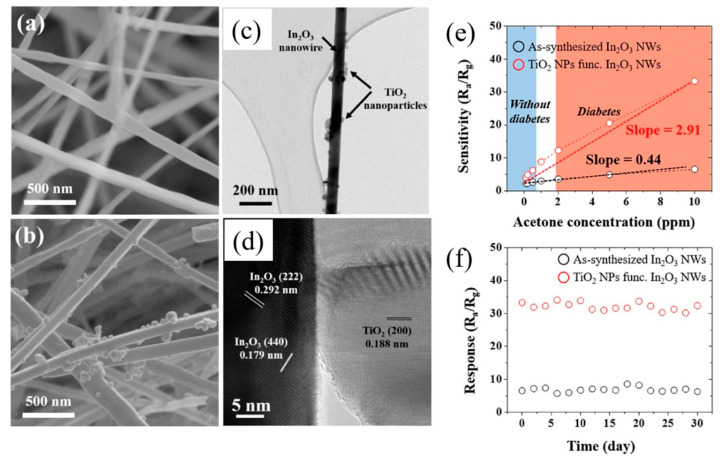
SEM micrographs of (**a**) pristine and (**b**) TiO_2_-decorated In_2_O_3_ NWs. (**c**,**d**) TEM images of TiO_2_-decorated In_2_O_3_ NWs. (**e**) Calibration curves and (**f**) long-term stability at 250 °C [90] (Used with permission from Elsevier^®^).

**Figure 16 materials-16-06233-f016:**
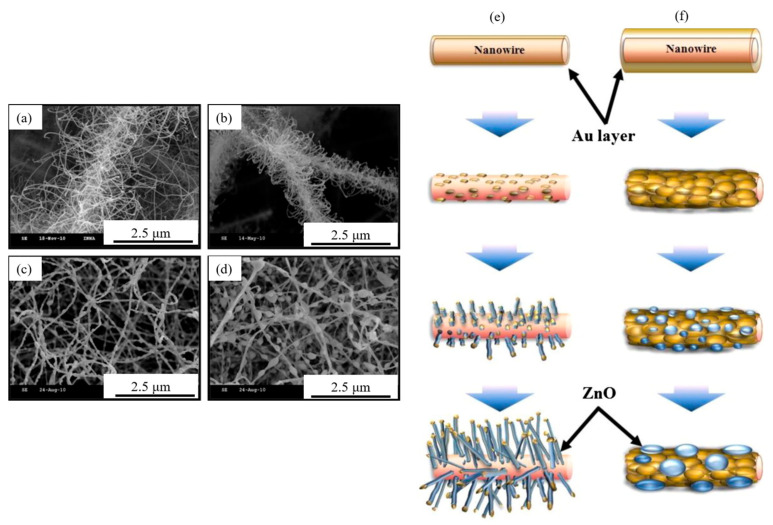
SEM images of ZnO-branched SnO_2_ NWs using initial Au thicknesses of (**a**) 1, (**b**) 3, (**c**) 10, and (**d**) 30 nm. Schematic of the synthesis of ZnO branches with Au having thicknesses of (**e**) 3 and (**f**) 10 nm [94] (Used with permission from Elsevier^®^).

**Figure 17 materials-16-06233-f017:**
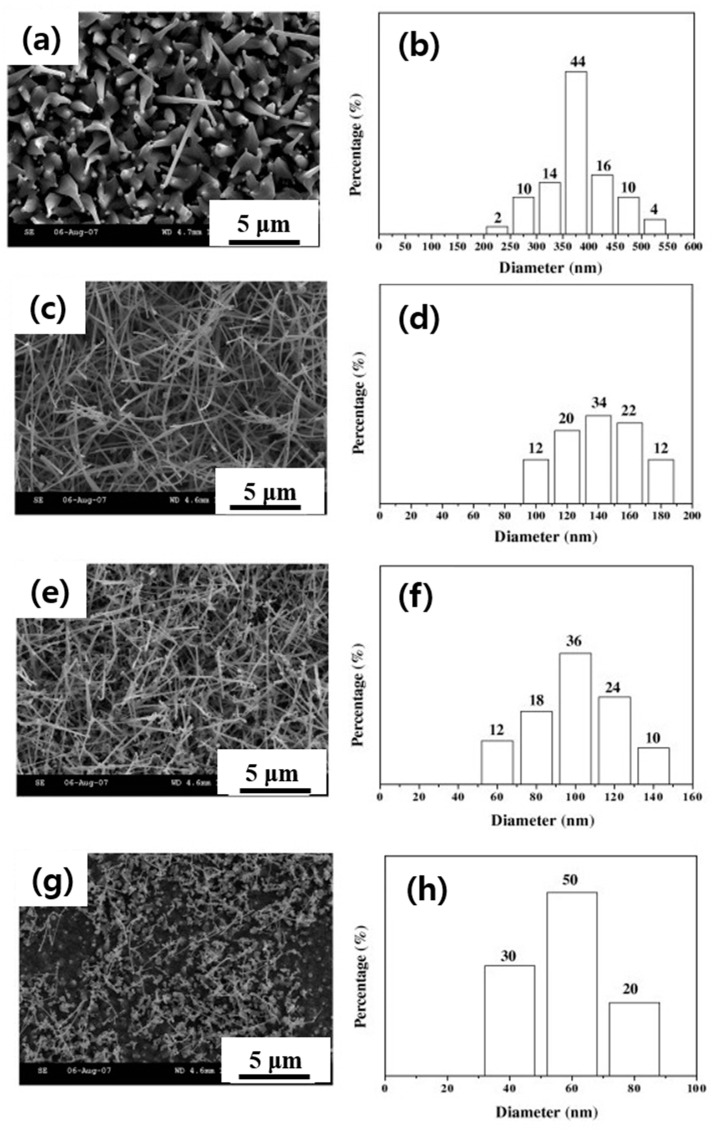
SEM images and diameter distributions of the ZnO NWs synthesized at (**a**,**b**) 900, (**c**,**d**) 950, (**e**,**f**) 1000, and (**g**,**h**) 1050 °C [95]. (Used with permission from Elsevier^®^.)

**Figure 18 materials-16-06233-f018:**
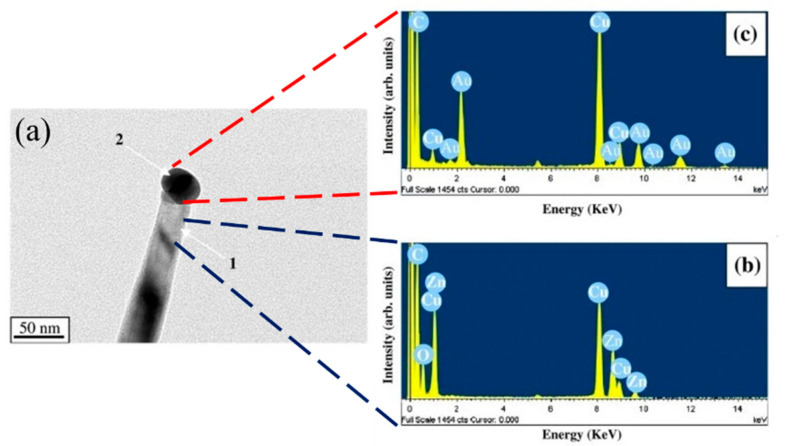
(**a**) TEM image of a single ZnO NW grown at 950 °C. (**b**,**c**) TEM-EDS spectra taken from different areas in (**a**) [95] (Used with permission from Elsevier^®^).

**Figure 19 materials-16-06233-f019:**
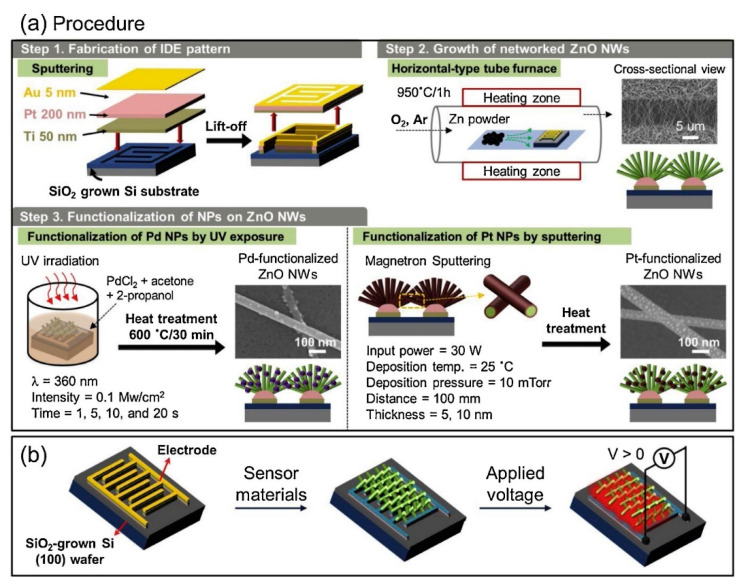
(**a**) Synthesis procedure of Pt and Pd-decorated ZnO NWs and (**b**) fabrication of a sensor and the self-heating mode of operation [96] (Used with permission from Elsevier^®^).

**Figure 20 materials-16-06233-f020:**
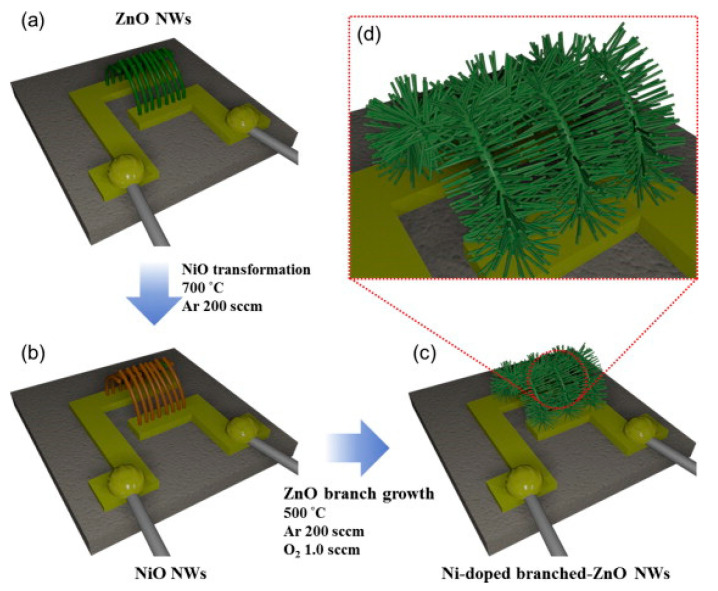
(**a**) Growth of ZnO NWs, (**b**) transformation into NiO NWs, and (**c**,**d**) growth of Ni-doped ZnO branches [97] (Used with permission from Elsevier^®^).

**Figure 21 materials-16-06233-f021:**
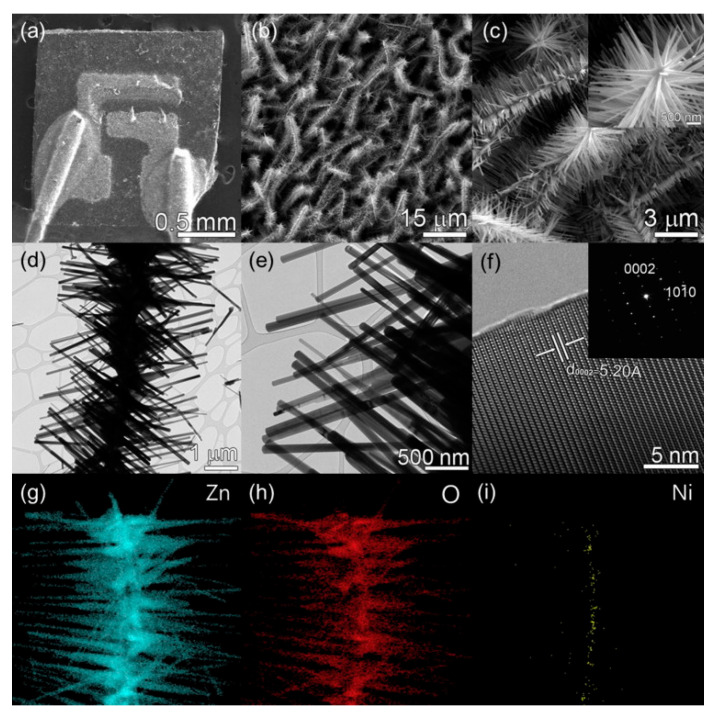
(**a**–**c**) SEM, (**d**,**e**) TEM, and (**f**) HRTEM images of Ni-doped branched ZnO NWs and SAED pattern (inset). (**g**–**i**) EDS mapping [97] (Used with permission from Elsevier^®^).

**Figure 22 materials-16-06233-f022:**
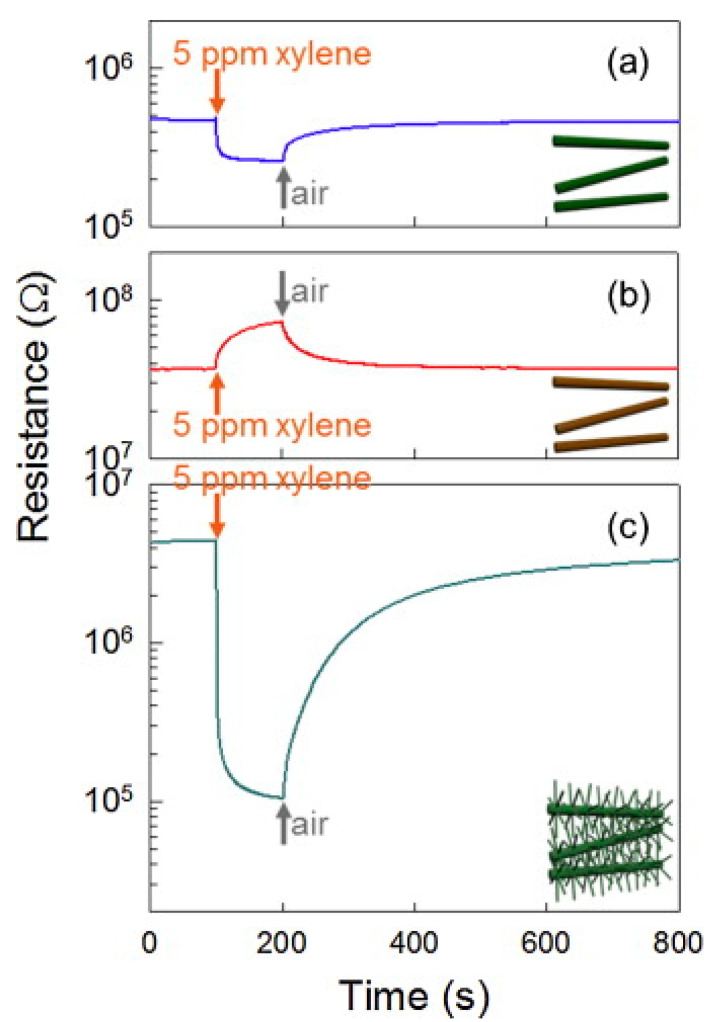
Dynamic resistance curves to 5 ppm p-xylene at 400 °C for (**a**) ZnO NWs, (**b**) NiO NWs, and (**c**) Ni-doped branched ZnO NWs [98] (Used with permission from Elsevier^®^).

**Figure 23 materials-16-06233-f023:**
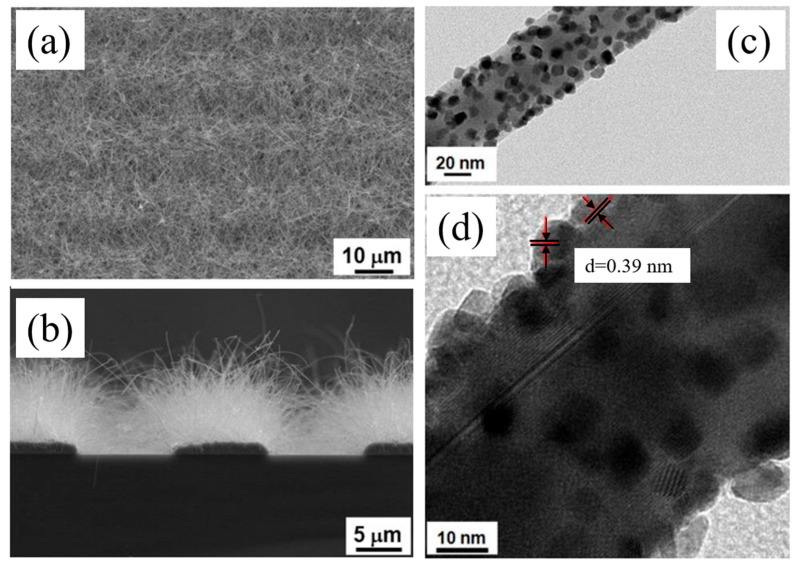
(**a**,**b**) FE-SEM images of ZnO NWs grown on a patterned electrode. (**c**) TEM and (**d**) HRTEM images of Pd-decorated ZnO NWs [99] (Used with permission from Elsevier^®^).

**Figure 24 materials-16-06233-f024:**
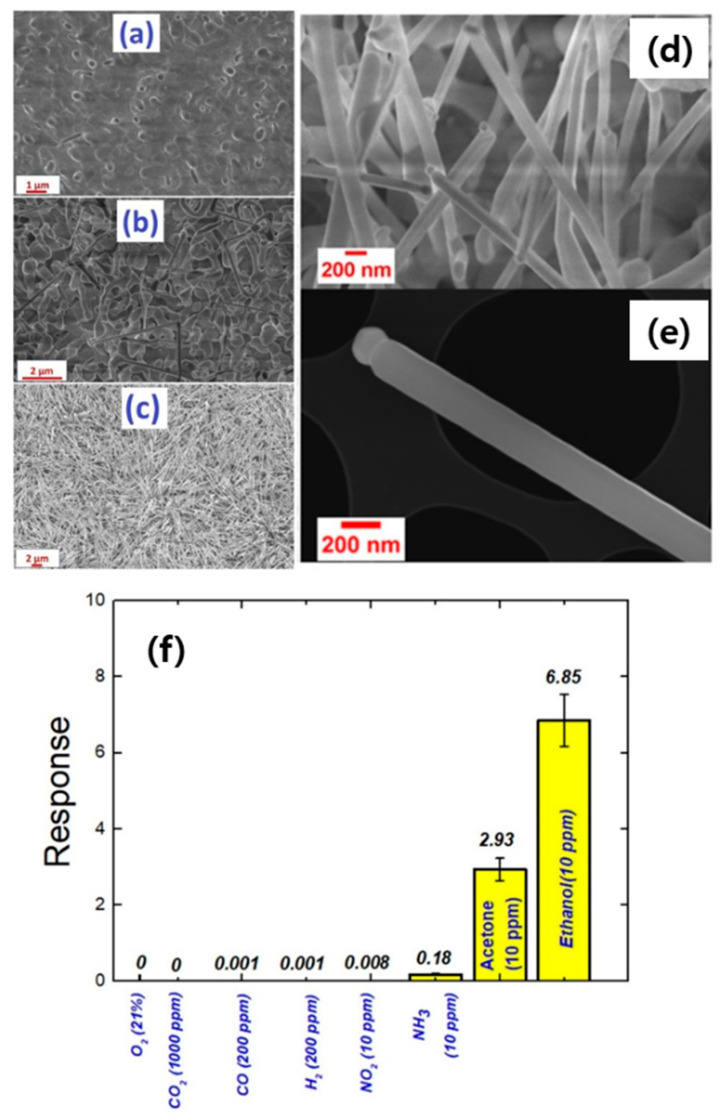
Bi_2_O_3_ materials grown using (**a**) Cu, (**b**) Pt, and (**c**) Au catalysts. SEM images of (**d**) Bi_2_O_3_ NWs and (**e**) a single Bi_2_O_3_ NW. (**f**) Selectivity pattern of a Bi_2_O_3_ NW sensor at 350 °C [104] (Used with permission from Elsevier^®^).

**Table 1 materials-16-06233-t001:** Gas-sensing properties of VLS-grown sensing materials.

Sensing Material	Gas	Conc. (ppm)	T (°C)	Response (R_a_/R_g_) or (R_g_/R_a_)	LOD (ppm)	t_res_(s)/t_rec_(s)	Selectivity Ratio	Ref.
SnO_2_ NWs	H_2_	1000	100	12 (ΔR/R_g_)	100	~180/~180	-	[55]
SnO_2_ NWs	O_2_	1 torr	RT	30% [(ΔR/R_g_) × 100]	0.05 torr	~200/~250	-	[56]
SnO_2_ hierarchical NWs	LPG	2000	350	20	500	~4/~8	-	[59]
SnO_2_ hierarchical NWs	NH_3_	300	200	10.9	300	~16/~25	-	[59]
SnO_2_ NWs	NO_2_	100	300	25	5	~600/~700	-	[60]
In_2_O_3_ NWs	CO	10	300	1.4	10	~240/~460	-	[61]
SnO_2_ NWs	NO_2_	10	200	43%	1	~38/~25	-	[62]
SnO_2_ NWs	NO_2_	0.5	200	20	0.5	43/18	-	[63]
SnO_2_ NWs	H_2_	1000	150	6 (ΔR/R_g_)	100	-/-	-	[64]
Co_3_O_4_-decorated SnO_2_ NWs	C_6_H_3_O	50	300	63	0.5	10/~300	2.2	[65]
Fe_2_O_3_-decorated SnO_2_ NWs	C_2_H_5_OH	200	300	60	0.5	~60/~110	1.58	[66]
CuO-decorated SnO_2_ NWs	H_2_S	1	300	5	1	~300/~400	-	[68]
Porous Si/SnO_2_ NWs	H_2_S	20	100	2	10	~200/~230	1.8	[69]
Pd-decorated SnO_2_ NWs	NO_2_	30	300	1.56	0.1	~470/~460	-	[70]
Pd-decorated SnO_2_ NWs	H_2_	1	300	8	0.1	~250/~300	8	[71]
Pd-decorated SnO_2_ NWs	NO_2_	2	25	6	2	~380/~1000	9.8	[72]
In_2_O_3_ NWs	C_3_H_6_O	500	Heat@ 1.06 mW	1.3	10	10/50	1.2	[83]
NiO foam@Sn-doped In_2_O_3_ NWs	Ethylene glycol	100	125	170	1	~9/~20	16	[85]
ZnO-doped In_2_O_3_ NWs	C_2_H_5_OH	100	200	62	-	-/-	2	[86]
In_2_O_3_ NWs	H_2_	500	200	0.7 (ΔR/R_g_)	500	31/80	-	[88]
Pt-decorated In_2_O_3_ NWs	O_2_	400	50	1.5	10	~70/~500	-	[89]
TiO_2_ decorated In_2_O_3_ NWs	C_2_H_5_OH	10	250	34	0.10	~15/~130	~2	[90]
Pd-decorated In_2_O_3_ NWs	NO_2_	3	50	3.5 (ΔR/R_g_)	3	60/365	-	[91]
Fe_2_O_3_-decorated ZnO NWs	CO	100	300	19	1	~100/~100	3	[93]
ZnO-branched SnO_2_ NWs	NO_2_	100	250	6	0.5	~1000/~12,000	-	[94]
Ni-doped branched ZnO NWs	Xylene	5	400	42	0.25	~80/~400	1.9	[97]
Pd-decorated ZnO NWs	CO	0.1	25	1.02	0.1	210/400	-	[99]

Note: LOD = limit of detection; t_res_/t_rec_: response time/recovery time; selectivity ratio: the highest response to target gas divide to the highest response to an interfering gas. (-): no data available.

## Data Availability

These data can be found only in this article.

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
