# Peer review of "Metal Oxide Nanowires Grown by a Vapor–Liquid–Solid Growth Mechanism for Resistive Gas-Sensing Applications: An Overview"

_materials, 2023, doi:10.3390/ma16186233_

Round 1
Reviewer 1 Report
This review presents a survey on metal oxide nanowires (SnO2, In2O3, ZnO, Bi2O3, WO3, NiO, Ga2O3, Fe2O3, etc.) grown by a vapor-liquid-solid growth mechanism for resistive gas sensing applications. The study is well organized and the readers can be interested on results. However, Minor Revisions are suggested:
- a table of comparison of the gas sensor performance is missed. Please, add a table to appreciate the gas sensing characteristics of MOX NWs in terms of sensitivity, limit of detection, cross-sensitivity, sensor temperature, etc.
After this revision, the manuscript could be publishable.
Minor editing of English should improve the manuscript
Author Response
Response is uploaded as a separate word file.

Reviewer 2 Report
The authors have done an interesting study. The following comments need to answered for acceptance of the review paper.
1) Title of the paper should be made clear so the reader would understand that it is a review paper.
2) Many review papers have been published on metal oxide nanowires with VLS mechanism. The need of the current review paper has to be stated and explained well in the introduction.
3) The findings have to be analyzed with respect to the aim of the review paper.
N/A
Author Response

(The authors gave the same response as above.)

Reviewer 3 Report
The authors have reviewed the VLS grown metal oxide nanowires, their morphologies and gas sensing properties. Basically, the authors could give a comprehensive introduction on the VLS, especially on the growth mechanism and controlling factors. A lot of references are summarized. In my opinion, this work could be published when the authors could address the following comments.
1. Metal oxide nanowires for gas sensing application is a relatively old topic. Nanowires have advantages for gas sensing, however, honestly, they also have severe limitations for gas sensing. That is why there are much less gas sensing papers in comparation with nanoparticles sensors. And all the commercial sensors, i.e., Figaro, Fis, Sensirion, employ oxide (mainly SnO2 based) nanoparticles based sensors. The authors should emphasize the unique advantages of nanowire based sensor. And especially recent papers (i.e. within recent 5 years). In this case, the review will become meaningful.
2. The authors should carefully check the writing. For example, the first sentence in introduction: “nanowires (NWs) have two dimensions in the range of 1-100 micrometers”, it should be “nanometer”.
3. Some of the description is not exactly accurate. For example, in abstract, “possibility of doping”, almost all the nanowire synthesis methods allow to dop. “good control over the synthesis parameters including the growth temperature, time, and pressure.”, this sentence is less meaningful, any synthesis method has its own control parameters.
Basically ok, but can be further improved
Author Response

(The authors gave the same response as above.)

Round 2
Reviewer 3 Report
Revision is quite fast but inadequate.
The real advantage of naowires for gas sensing is still missing.
Revision is quite fast but is inadequate.
Author Response
The reply to the reviewer's comments is uploaded as a separate file.
